# A Comprehensive Review of the Nutritional Composition and Toxicological Profile of Date Seed Coffee (*Phoenix dactylifera*)

Raphaela Kiesler [1,2], Heike Franke [1] and Dirk W. Lachenmeier [2,*]

1  Postgraduate Study of Toxicology and Environmental Protection, Rudolf-Boehm-Institut für Pharmakologie und Toxikologie, Universität Leipzig, Härtelstraße 16-18, 04107 Leipzig, Germany; eq07xino@uni-leipzig.de (R.K.); heike.franke@medizin.uni-leipzig.de (H.F.)
2  Chemisches und Veterinäruntersuchungsamt (CVUA) Karlsruhe, Weissenburger Strasse 3, 76187 Karlsruhe, Germany
*  Correspondence: lachenmeier@web.de; Tel.: +49-721-926-5434

**Featured Application: This review evaluates the potential of date seed coffee as a sustainable and caffeine-free alternative to traditional coffee. The authors propose an environmentally friendly approach to beverage production by utilizing discarded date seeds. The findings provide comprehensive evidence supporting the safe consumption of date seed coffee, as demonstrated by a thorough toxicological risk assessment. This application aims to reduce waste and introduce a new culinary use for date seeds, providing a distinctive flavor profile and potential health benefits to the European market.**

**Abstract:** Approximately 8 million tons of dates (*Phoenix dactylifera*) are produced globally each year. The seeds of the fruit, which make up 10–15% of its weight, are typically discarded. Date seed coffee is a sustainable food system innovation rooted in the traditions of high date-producing regions. Dating back to the late 19th century, date seed coffee has evolved from a historical coffee substitute to a modern caffeine-free alternative. Date seed coffee has a long history of consumption in the European Union (EU). This indicates that it may not require novel food authorization. The composition of date seeds is evaluated in this review and a toxicological risk assessment for date seed coffee is conducted. Subchronic studies show that consuming date seed or date seed coffee has no adverse effects. Therefore, currently unavailable chronic toxicity, carcinogenicity, and reproductive toxicity studies may be unnecessary. However, for a comprehensive evaluation, it is recommended to conduct an in vitro mutagenicity test. This review provides information on the safety of date seed coffee and highlights the need for further research.

**Keywords:** date seed coffee; *Phoenix dactylifera*; coffee by-products; date by-products; risk assessment; toxicity

## 1. Introduction

Date trees play a significant role in the daily lives of many people [1]. The fruits provide nutrition for both humans and animals [2]. Worldwide, approximately 8 million tons of date fruits are produced annually, with 90% of the global production distributed among the top ten date-producing countries, led by Egypt, Saudi Arabia, and Iran [3]. Date seeds are often removed and discarded, which results in a high amount of date seed waste. Traditionally date seeds are used as animal feed [4]. In recent years, various methods have been proposed for reusing date seeds in both the food and non-food industries [5]. The waste of date seeds has the potential to provide economic benefits to the date industry, as well as possible health and nutritional benefits [1,6]. Approximately 40% of all food waste in the food chain occurs during manufacturing. This review focuses on the interesting option of further processing discarded date seeds into a coffee-like product [7].

The date palm belongs botanically to the family Arecaceae. The date palm tree (*Phoenix dactylifera*) is one of the 12 species of the genus *Phoenix* and the most important for commercial cultivation and the food industry. The date palm fruit contains a single hard seed encased by the endocarp of the fruit flesh, followed by the mesocarp (pulp), and finally surrounded by the epicarp (skin) [8]. The kernels are reddish in color and enclosed in a seed coat [9,10] (see Figure 1). Date palms spread naturally through endochory, meaning that the date seeds must be able to withstand the mammalian and avian digestive systems.

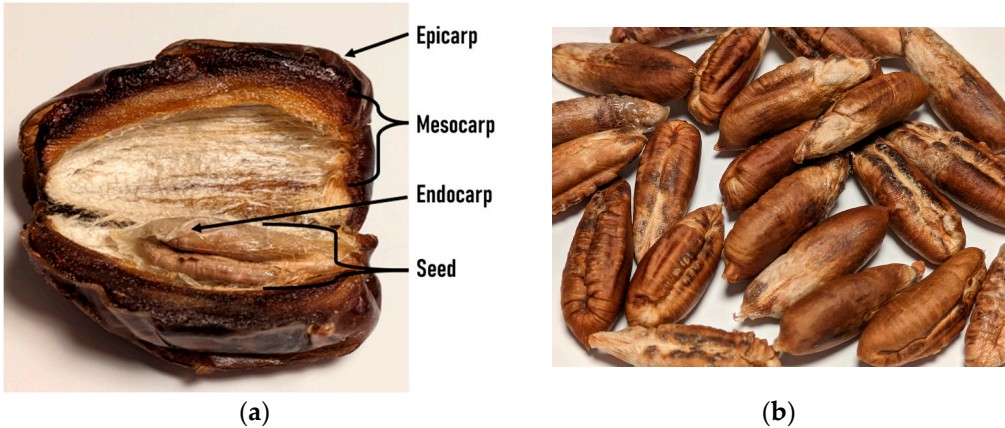

(**a**)                                                            (**b**)

**Figure 1.** Macroscopic pictures of (**a**) date fruit anatomy; (**b**) raw date seeds.

The composition of dates varies slightly, depending on the cultivar [11,12] and agroclimatic conditions [4,13]. Dates are typically harvested after a natural loss of moisture and an increase in the sugar content. This natural drying process occurs while the fruit is still on the date palm [13,14]. Dates have a sweet and caramel-like flavor. The fully ripe stage of the date fruit maturation is called "Tamar" (see Figure 2). Date seeds constitute about 10–15% of the total fruit weight and have no noticeable flavor or scent in their unprocessed state [11,12]. There are over 2000 known date varieties worldwide.

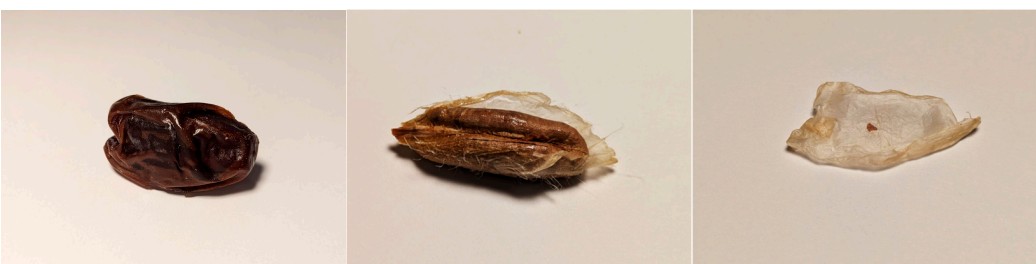

**Figure 2.** Macroscopic pictures of date seed components. From **left** to **right**: date, fully ripened "Tamar"; date seed encased with endocarp; endocarp.

The use of date seeds in coffee beverages has a long history, not only in countries where dates are grown, but also in the European Union. The first German references to date seed coffee date back to the 19th century, more specifically to the years 1881, 1886, and 1891. These records provide information on trade in southern Bavaria, an analysis of the date seeds, and the possibility of conducting microscopic examinations [8,15,16]. In 1918, date seed coffee was described in a German anthology on the study of food, edibles, and stimulants. The analytical studies are rudimentary, focusing on reflected-light microscopy and dry matter analysis. However, based on the citation in the book by Beythien et al., it can be assumed that date seed coffee was consumed regularly [9]. This is consistent with a corresponding entry in a handbook of pharmaceutical practice from the year 1927, wherein the use of date seed coffee is described [17]. From 1914 onwards, and especially during the years of the First World War, the use of coffee surrogates has considerably increased.

Date seeds were used as a substitute for coffee, and historically, they were also used as an admixture in spices to adulterate more expensive ingredients [18].

In more recent years, date seed coffee products were "reinvented" as a caffeine-free substitute, and also as a functional food. A German patent applied for a coffee-like recipe made from date seeds and cardamom in 1989 [19]. In the year 2011, a patent application was published in the USA for a roasted date seed beverage, including its composition, production, and beverage preparation [20]. Since 2017, a company in California, USA, has been offering date seed coffee products. As of 2023, a Dutch company is offering multiple date seed coffee products for trade within Europe. Another supplier based in Germany offers date seed coffee via online trading throughout Germany. Historical evidence indicates that date seed coffee may have been consumed to a significant degree by humans in the EU before 15 May 1997. Therefore, pre-market authorization as a novel food should not be necessary for date seed coffee. While the EU novel food catalogue does not currently include an entry on coffee from *Phoenix dactylifera*, the Dutch responsible authority has confirmed that date seed coffee is not novel [21].

The aim of this review article is to summarize the composition of date seeds and provide a toxicological risk assessment for date seed coffee based on the available literature, with a focus on EU food safety requirements. On the one hand, the dates are considered in their raw state, but also as a roasted product, which, in most cases, will be the starting point for date seed coffee.

## 2. Materials and Methods

Through electronic search functions, the databases PubMed (National Library of Medicine, Bethesda, MD, USA) and Google Scholar (Google LLC, Mountain View, CA, USA) were searched for the keywords "date seeds", "date coffee", "date seeds toxicity", "date seeds toxic", "date seeds mycotoxin", "date seeds acrylamide", and "date seeds phytosterol". The Google Scholar search was also conducted using French and German translations of the keywords.

The focus of the regulatory toxicological assessment was regarding food safety criteria in the European Union. Therefore, information from the European Food Safety Authority (EFSA), the German Federal Institute for Risk Assessment (BfR), and the Austrian Agency for Health and Food Safety GmbH (AGES) were used to obtain general information on nutrition and food intake, for published risk assessments, and scientific opinions on food constituents. The literature research was conducted in August 2023.

The obtained references were evaluated after assessing the abstract and full text to make sure that the presented study showed suitable results for this research topic. The bibliographies in the references were also checked for additional publications for the respective research topic, and authors with relevant publications were further screened using Google Scholar (Google LLC, Mountain View, CA, USA) and Research Gate (ResearchGate GmbH, Berlin, Germany) to check their list of publications.

Some references were only available in a non-English language (e.g., Arabic, French, and German). In these cases, DeepL Translator (DeepL SE, Cologne, Germany) was used for translation.

## 3. Compositional Data on Date Seeds and Date Seed Coffee Products

The methodology for processing date seeds to create a coffee-like beverage closely parallels the traditional preparation of coffee beans from *Coffea arabica* or *Coffea canephora*. The initial step involves the separation of the seed from the surrounding pulp, which is critical for ensuring the purity of the product. Subsequent to this, the seeds undergo a series of processes: immersion and cleansing in water to remove any adherent contaminants, dehydration, and roasting at a predetermined temperature to achieve the desired flavor profile. The roasted seeds are then ground to a specific coarseness, akin to coffee grounds, and subjected to a hot water extraction process, mirroring the conventional brewing of

coffee. This study aims to evaluate the date seeds both in their raw state and post-roasting to ascertain any changes in their compositional attributes attributable to the roasting process.

### 3.1. Macronutrients

Macronutrients are essential calorie-containing components for human nutrition and growth [22]. A rough segmentation is made into carbohydrate, fat, and protein. In Table 1, an overview of the compositional data for date seeds is given. The nutritional breakdown is provided for various types of dates, both raw and processed. One of the earliest descriptions of the date seed composition in Germany is from 1918, and although the analytical technique is not described, the results are consistent with later studies [9].

#### 3.1.1. Protein

The protein content of the dry seed matter is only 5%, and the ash content is approximately 1%. Carbohydrates make up the majority of the seed, accounting for over 80% of its composition (examples are given in Table 1; 83% *Deglet Nour* and 81% *Allig* are well-known Tunisian varieties) [11]. Similar values were detected by Nehdi et al. for the species *P. canariensis* [23]. At certain temperatures, the protein content noticeably decreases. For instance, roasting the sample for over 20 min at 220 °C reduces the protein content to 8.8%, whereas the untreated sample has a protein content of 11.1% (14.1% at 180 °C; 13.3% at 200 °C) [24]. In addition, it must be considered that the centrifugation technique, the liquid ratio of the test substance, and the extraction time could have an impact on the concentration of the protein content, besides temperature differences [25].

#### 3.1.2. Carbohydrates

Compared to date flesh, date seeds only contain a small amount of soluble sugars, but a rather high amount of fiber [26–28]. A total of 16–35% of the total mass consists of glucose, fructose, raffinose stachyose, sucrose, and galactose [26].

**Table 1.** Macronutrients of date seeds, pure and roasted (g/100 g [1] or % fresh weight/d.w. [2] or roasted [3]).

| Material/Date Variety | Raw/Roasted | Carbohydrates (Fiber) | Ash | Fat | Protein | Reference |
|---|---|---|---|---|---|---|
| Date seeds [2,3] | Roasted | 84.3 (29.7) | 1.4 | 8.5 | 5.8 | [9] |
| *Hillawey* (Iraq) [1] | Raw | 80.5 (29.1) | 1.5 | 5.7 | 6.4 | [29] |
| *Saidy* (Egypt) [2] | Raw | 73.6 | 3.4 | 9.6 | 6.2 | [25] |
| *Degla-Baïdha and Tafezouine* [1] | Raw | - | - | 5.5 | - | [30] |
| *Deglet Nour, Allig* (Tunisia) [2] | Dried at 50 °C | 83.1; 81.0 | 1.1 | 10.2; 12.7 | 5.6; 5.2 | [11] |
| Summary: 23 date varieties [1] | n.a. | 70.9–86.9 | 0.9–1.2 | 5.0–12.5 | 2.3–6.9 | [31] |
| *Red Sayer* (Emirates) [2,3] | Roasted at 125 °C for 30 min | 78.5 | 1.2 | 7.3 | 8.6 | [32] |
| *Kabkab* (Iran) [1] | Raw; Date seed flour | 77.3 (22.1) | 1.1 | 9.6 | 5.5 | [33] |
| *Soukari* (Saudi Arabia) [1,3] | Raw; Roasted at 220 °C | - | 0.9; 1.0 | 8.6; 9.8 | 11.1; 8.8 | [24] |
| *P. canariensis* (Emirates) [2] | Dried at 60 °C for 24 h | 72.6 | 1.2 | 10.4 | 5.7 | [23] |
| Roasted seeds (Oman) [1,3] | Roasted at 220 °C for 15–20 min | 83.7 (21.4) | 1.0 | 8.1 | 7.1 | [2] |
| Roasted date seeds (commodity: India) [1,3] | Roasted | 76 | 8.8 | 0.3 | 11.6 | [34] |
| Roasted date seeds (Jordan) [2,3] | Roasted | 60.8 (6.7) | 17.2 | 0.8 | 11.7 | [35] |
| Heated date seed powder (Tunisia) [2] | Oven- and sun-dried | 25.6; 33.8 | - | 17.4; 31.5 | 15; 26.5 | [36] |

Different heating techniques, and heating in general, can affect the levels of macronutrients. The carbohydrate, fat, and protein values are lower after heating in the oven than after solar thermal treatment [28].

### 3.1.3. Fiber

Dietary fiber is defined as non-digestible carbohydrates, such as pectin, cellulose, hemicellulose, and different types of non-starch and resistant saccharides, as well as lignin. The carbohydrates found in date seeds are primarily insoluble fiber types such as cellulose and hemicellulose. Two-thirds of the fiber content in the Tunisian date cultivar *Deglet Nour* is attributed to cellulose, while 50% and 20% are attributed to hemicellulose. Around 70% of the components of *Allig* date seeds are fiber, with slightly higher cellulose content and lower hemicellulose content. The total fiber content of *Deglet Nour* and *Allig* is complemented by a low lignin content [37]. In the endosperm, there are various hemicellulose fractions, including gluco and galacto-mannan, and alkali-soluble heteroxylan. These storage compounds are slowly degraded and support the date seeds during the long germination period [13]. Al-Farsi et al. reported an insoluble fiber content of 52.7%. After phenolic extraction, either with solvent water or acetone, the fiber extraction increased to 81–82 g/100 g [37]. A daily fiber intake of 25 g per day is recommended for adults to support normal laxation. In addition, other positive health effects of dietary fiber are associated with reduced risk of heart diseases and the prevention of diet-related diseases [38]. Date seed fiber is a good source of fiber for human nutrition and can be easily added to bread doughs. In summary, dates are good sources of carbohydrates and fiber, with a lower sugar content.

### 3.1.4. Fat

Besbes et al. described in detail the chemical composition of two studied date cultivars, *Allig* and *Deglet Nour* [11]. The fat content corresponds to about one-eighth of the whole date seed, with 10.2% fat content in *Deglet Nour* and 12.7% in *Allig*. Lower fat content was measured for the cultivars *Degla-Baïdha* and *Tafezouine*, with around a 5.5% fat total [1,30].

Date seeds typically contain between 5 and 13% fat, with the distribution of fatty acids being influenced by the date variety [31], growing conditions, and extraction technique used. The oil content of the Tunisian cultivars (*Allig* and *Deglet Nour*) is comparatively high. Hexane is widely used as a solvent and the Soxhlet method is commonly employed for extracting date seed lipids. Gas chromatography (GC) was performed to determine the detailed fatty acid composition, although non-conventional methods such as UAE and SC-$CO_2$ can be used as environmentally friendly alternatives [1,39].

An overview of the oil contents and fatty acid composition of almost 20 different important date varieties is provided in Table 2. About 92% to 99% of the fatty acid content is distributed among the most important acids, including: oleic, linoleic, palmitic, myristic, lauric, and minor stearic [11,23]. A similar fatty acid composition can be found in commonly used olive oil or other edible oils [40]. The oxidative stability of date seed oil is high and comparable to that of olive oil [11]. The tested date seeds have the highest concentration of oleic acid. Date seed oil has similar characteristics to edible vegetable oil; thus, it can be considered safe for human consumption [6].

Table 2 summarizes the investigated fatty acid contents. Where specified, the date variety is mentioned. Bigger differences are indicated for the lauric acid content in a roasted date seed sample, although the other results are quite homogeneous. The difference between fatty acid concentrations is linked to the date varieties, roasting processes, and pollination [2]. Slight increases and decreases in the fatty acid content may occur after the roasting processes of the date seed powder [24]. The roasted samples contain values that are two to three times higher than those found in the comparison group, which only contains unprocessed date seeds [41]. Most varieties have a comparatively low proportion of fat and protein, which amounts to only 5–10% of the total composition. The fatty acid composition may vary slightly between cultivars, but they exhibit good oxidative stability and a high content of unsaturated acids.

**Table 2.** Fatty acid composition of date seed oils (%).

| Fatty Acids | | Composition (Range in %) | Reference |
|---|---|---|---|
| Oleic acid, $C_{18:1}$ | Monounsaturated | 39.2–50.1 | [2,11,23,24,26,30] |
| Lauric acid, $C_{12:0}$ | | 5.8–38.8 | [2,11,23,24,26,30] |
| Myristic acid, $C_{14:0}$ | Saturated fatty acid | 3.1–12.8 | [11,23,24,26,30] |
| Palmitic acid, $C_{16:0}$ | | 7.8–15.1 | [2,11,23,24,26,30] |
| Stearic acid, $C_{18:0}$ | | 1.3–5.7 | [11,23,24,26,30] |
| Linoleic acid, $C_{18:2}$ | Polyunsaturated | 6.1–21.0 | [2,11,23,24,26,30] |
| Linolenic acid, $C_{18:3}$ | | 0.1–2.3 | [11,23,24,26,30] |

*3.2. Minerals*

Several studies have analyzed the mineral composition of date seeds. Table 3 provides an overview of the mineral contents of different sources. Considering that date seeds are used for coffee surrogate preparation and consumed daily in the same recommended intake levels as coffee, generally, no health risk is expected. The mineral intake concentration is within the reference quantity. Compared to coffee and barley-coffee drinks, date seed coffee has the lowest mineral content. The mineral content of date seed coffee is only slightly higher in sodium, iron, and zinc compared to *Coffea arabica* coffee, but still within the recommended dietary intake level [42]. A high intake of the trace element nickel occurs through the consumption of coffee. However, since date seeds contain less nickel than *C. arabica* coffee, substituting the daily coffee beverage intake with products made from date seeds does not pose any health risks [43]. Additionally, date seeds are a good source of calcium and potassium [23].

Mineral contents may vary more between varieties, but, overall, it is unlikely that the recommended daily intake will be exceeded.

**Table 3.** Mineral content of commercially roasted date seed powders (mg/kg).

| Minerals | [41] | Amount (Range in mg/kg) | Reference | Maximum Intake Levels |
|---|---|---|---|---|
| Sodium (Na) | *Main elements and minerals* | 8.8–268.7 | [2,4,11,23,24,42,44] | n.a. [45] |
| Potassium (K) | | 21.9–4857.6 | [2,4,10,11,23,24,42,44] | n.a. [45] |
| Calcium (Ca) | | 27.1–1472 | [2,4,10,11,23,24,42,44] | 2500 mg/d * [45] |
| Magnesium (Mg) | | 10–788.5 | [10,11,23,24,42,44] | 250 mg/d [45] |
| Copper (Cu) | *Trace elements* | 5.2–8.7 | [2,4,10,24,42] | 5 mg/d [45] |
| Iron (Fe) | | 0.9–150.5 | [2,4,10,11,23,24,42,44] | n.a. [45] |
| Manganese (Mn) | | 6.2–17.9 | [2,4,10,24,42] | n.a. [45] |
| Zinc (Zn) | | 0.7–18.4 | [2,4,10,24,42,44] | 25 mg/d [45] |
| Chromium (Cr) | | <0.05–9.7 | [2,4,24] | n.a. [45] |
| Nickel (Ni) | | 1.12 | [42] | 13 µg/kg/d (TDI) [42] |
| Tin (Sn) | *Ultra-trace elements* | 6.43 | [35] | 250 mg/kg [46] |
| Aluminum (Al) | | 5–21.2 | [4] | 1 mg/kg b.w. (TWI) [47] |
| Arsenic (Ar) | | 0.017 | [35] | 0.030 mg/kg ** [48] |
| Antimony (Sb) | | 0.059 | [35] | Negligible risk [49] |
| Mercury (Hg) | | 0.026 | [35] | 4 µg/kg b.w. (TWI) [50] |
| Cadmium (Cd) | | n.a., <0.05–0.42 | [2,4,10,35,42,51] | 2.5 µg/kg b.w. (TWI) [52] |
| Lead (Pb) | | n.a., <0.05–1.1 | [2,4,10,35,42,51] | 0.2 mg/kg (wheat) [48] |

* Upper level; ** Non-alcoholic rice-based drinks; Not applicable (n.a.).

*3.3. Vitamins*

Vitamin E is a group of lipophilic antioxidants, known as tocopherols (four isomers, α, β, γ, and δ), that are naturally formed in plants. They are essential for supporting several endogenous functions in human health, such as the immune and nervous systems, the protection of vessels, and general antioxidant effects. The recommended daily intake is 11 to 15 mg of vitamin E. An additional maximum intake of 30 mg/d through supple-

ments should not be exceeded [53]. According to the EFSA's tolerable upper intake level, up to 300 mg/d of vitamin E does not cause harm to human health [45]. Besides vitamin E, certain amounts of vitamin $K_{1-3}$ are also present [44].

The main detected tocopherols in date seed oils from different cultivars were $\alpha$-tocotrienol (31.8–37.4 mg/100 g oil), $\gamma$-tocopherol (7.6–11.8 mg/100 g oil), and $\gamma$-tocotrienol (4.6–8.5 mg/100 g oil), which were followed by $\delta$-tocopherol (1.1–2.8 mg/100 g oil) and $\beta$-tocopherol (0.7–1.3 mg/100 g oil). A total of around 51.5 mg tocols (tocopherols and tocotrienols) in 100 g oil were measured [23,26]. $\alpha$-Tocopherol is known for its high biological vitamin E activity, which is 10 to 30% higher than $\gamma$-tocopherol. An $\alpha$-tocopherol content from 0.3 to 1.7 mg/100 g oil was detected [23,26,36].

### 3.4. Plant Secondary Metabolites

3.4.1. Phenolic Acids and Flavonoids

Date seeds contain a relevant amount of phenolics, primarily composed of flavonoids, which are known for their high antioxidant capacity [5]. Among the different components of the date fruit, the seeds exhibit the highest antioxidant activity, with an $IC_{50}$ value of approximately 24 μg/mL. Date seed coffee, which has a very strong antioxidant activity [32], is linked to its high phenolic content [26,37]. Phenolic compounds preserve fruits against microbial and parasite infections, as well as photo-oxidation. The seeds of a plant are essential for its survival, and the high antioxidant content in the seeds may be due to their high protection value [13]. Date seeds have been found to have an inhibitory effect against Gram-positive bacteria (*Staphylococcus aureus*), but no antibacterial activity has been reported against Gram-negative bacteria (*Escherichia coli*) [54]. The phenolic content can be affected by several production-specific factors, such as the cultivar, fruit maturity, growing conditions, season, and soil type [26]. Additionally, the final measured results may be skewed by the choice of extraction conditions [37].

The total flavonoid content is lower than the phenolic acid content. Approximately 1–2% of the total phenolic content (TPC) is flavonoids and contributes proportionally to the antioxidant activity. The *Deglet Nour* variety contains the highest amount of TPC (4166 mg gallic acid equivalent (GAE)/100 g d.w.) and the *Medjool* variety contains only about half of that amount [55]. A similar result was observed when Iranian date varieties were compared. The TPC ranged from 1483 to 3377 mg GAE per 100 g d.w. Seven primary polyphenols were identified, including: cinnamic, chlorogenic, caffeic, vanillic, gallic, 2,5-, and 3,5-dihydroxybenzoic acids [54]. In the *Bouhattam* roasted date seeds, extracted with dichloromethane, three different flavonoids were present with a total concentration of 4.1 mg/100 g, including: catechin, naringenin, and apigenin. Protocatechuic and quinic acids are the major compounds in *C. arabica* coffee beans and are also found in roasted date seeds [44]. Catechin, epicatechin, quinic acid, and protocatechuic acid are the main compounds, which are known for their high antioxidant activity. With the use of GC-MS, 13 different organic compounds were identified [44], as can be seen in Table 4.

Procyanidin A2, B1, and B2 were detected in date seeds of the *Deglet Nour* cultivar using the free polyphenol fraction extraction method [56]. It has also been shown that procyanidins from date seeds undergo digestion mainly unchanged. This could result in a positive effect on the intestinal microbiome [57]. The flavonoids luteolin and quercetin are present in date seeds roasted at 160 °C and 180 °C. A higher amount was detected in the sample roasted at higher temperatures due to the increased solubilization [1]. When comparing date seeds roasted at 160 °C, 180 °C, and 200 °C, the TPC increases with the roasting temperature and the browning of the date seeds. The higher TPC content can be attributed to chemical reactions occurring during the roasting process, which make free phenolic compounds more readily available. The hybridizability of the tannins and gallic acid was affected by the roasting and grinding of the date seed powder [58]. The phenolic content and oxygen radical scavenging capacity (ORAC) of the tested date seed beverages are influenced by the type of coffee preparation. The sample from the espresso machine contained the highest amount of gallic acid equivalents, followed by hot water suspension,

and lastly the filter brew drink (10.5 > 4.5 > 2.28 mM). The amount of coffee powder used for the espresso preparation was the lowest [59].

**Table 4.** Phenolic and volatile compounds in *Bouhattam* roasted date seeds (LC-MS and GC-MS) [44].

| Test Extracts (Increasing Polarity) | TPC (mg GAE/g Dry wt) | Antioxidant Activity (%) | Organic Compounds: Acids, Aldehydes, Alkanes, Phenols, and Sterols | |
|---|---|---|---|---|
| | | | LC-MS (12) | GC-MS (13) |
| Cyclohexane | n.d. | 1.4 | - | Undecane, 2-decenal, 2,4-decadienane, 6-methyl-octadecane, tetradecanoic acid, stearic acid, oleic acid, and β-sitosterol |
| Dichloromethane | 6.5 | 4.9 | Flavonoid only, 4.10 mg/100 g: Catechin (2 mg), naringenin and apigenin (with each 1 mg) | Phenol, stearic acid, n-hexadecanoic acid, 9-hexadecenoic acid, and oleic acid |
| Methanol | 56.7 | 94.4 | Three phenolic acids (quinic acid, protocatechuic acid, and caffeic acid); six flavonoids (catechin, epicatechin, quercetin-3-o-glucoside, luteolin-7-o-glucoside, naringenin, and luteolin) | Tetradecanoic acid, trans-13-octadecenoic acid, and stigmastan-3,5-diene |

### 3.4.2. Phytosterols

The analysis showed that date seed oil (*P. canariensis*) contained 336 mg/100 g of various phytosterols. The predominant sterol present was β-sitosterol, accounting for over 76% of the total, with around a 20% split between campesterol, $\Delta^5$-avenasterol, $\Delta^{5,24}$-stigmastadienol (2.73%), and a small amount of cholesterol (0.42%), stigmasterol (1.09%), $\Delta^7$-stigmastenol (0.79%), and $\Delta^7$-avenasterol (1.18%) [23]. Date seeds contained moderate quantities (7.2%–9.6%) of sterols and triterpenes, with only β-sitosterol and cycloartenol being identified. This substance class was not detected in coffee beans [41].

### 3.4.3. Carotenoids

Date seed oil from the cultivar *P. canariensis* has a light-yellowish color. An amount of 5.5 mg of carotenoids were detected in 100 g of date seed oil, which is the reason for the coloration [23]. Heating slightly reduced the carotenoid content. β-Carotene was proportionally the most abundant carotenoid in date seed oil, with 1.2–2.7 mg/100 g [24]. Compared to *C. arabica* brews, the color of coffee beverages made from the date seed powder was brighter and of a reddish and yellowish color [60].

### 3.4.4. Caffeine

Date seeds are naturally free of any amounts of caffeine compared to coffee powder from coffee plants, which contains approximately 0.5–1.5% caffeine [61,62]. Coffee-like beverages made from date seeds are a good option for consumers who prefer the taste of coffee but want to enjoy it without caffeine. The analysis of roasted date seed coffee was conducted by using HPLC and a standard solution of 200 mg caffeine per liter. No traces of caffeine in the date seeds were found [2]. Further studies were performed, but caffeine could not be detected [10,32,34,35,44,60,63,64].

### *3.5. Contaminants*
#### 3.5.1. Metals

Only a few studies have been conducted on the contaminants in date seeds. Table 3 presents a summary of the available data on the mineral content. Heavy metals, like

cadmium, lead, tin, arsenic, and mercury, can be easily taken up by the plant during the growth phase.

Although the lead and cadmium concentrations were lower in the examined *Bahraini* date seeds than in coffee beans, Ghnimi and Almansoori detected higher cadmium contents in the tested roasted date seed powder samples compared to Arabian coffee [4]. Since cadmium is known for its toxicity, a TWI of 2.5 µg/kg b.w. should not be exceeded [52]. However, cadmium and lead contaminants were only detected in trace amounts, making it unlikely to pose a risk to human health [49,65]. In their study, Mohd Jamil et al. investigated the heavy metal content in roasted date seed powder [35]. The results of this study found that the arsenic, mercury, and antimony content in date seeds was higher than that in *C. arabica* coffee, while the cadmium, tin, and lead content was comparable. However, all measured values were within the maximum permitted levels.

### 3.5.2. Toxins (Alkaloids and Mycotoxins)

No alkaloids were detected in a comprehensive screening with four different roasted date seed samples [4].

Maximum levels (µg/kg) for different mycotoxins in date products are regulated, but specific levels for date seeds are not available [65]. Dried fruit is also regulated for aflatoxin levels. During the ripening stage, khalal (green, unripe fruit) and rutab (fruit is partially ripened) date fruits can be attacked by *Aspergillus flavus*, which can result in aflatoxin contamination. *Aspergillus* ssp. is responsible for most date infections [14]. Commercially available date fruits were found to contain ochratoxin A-forming fungal species, specifically *Aspergillus niger*. Out of the 36 tested isolates, 9 showed ochratoxigenic potential [66]. Additionally, larger quantities of enniatin-B, produced by *Fusarium* species, were detected in date fruits. The manufacturing process, from growth to storage, can affect the microbiological contamination of the collected samples. This contamination is determined by temperature and humidity [67,68]. Research indicates that seeds are generally less contaminated than fruits. Microbiological contamination of the fruits poses a significant health risk and can also result in financial losses due to fruit spoilage. In a study of 20 different date varieties, over 81% of the fruits were found to be contaminated with *Aspergillus niger*, 47% with *Aspergillus flavus*, and some with *Aspergillus* sp. While seeds are generally less contaminated, values ranging from 8% to 58% infested seeds were measured, and no cultivar was completely free of microbiological contamination [68]. Due to the detected values of microbiological contamination, it is important to test dates and date seeds before further use and processing. This is supported by a large screening study that compared the contamination levels of traded goods worldwide. Almaghrabi found that highly regulated markets, such as the European Union, have lower contamination levels compared to developing countries with less-well-regulated food systems [69].

When considering roasted date seeds, it may be appropriate to use the maximum level for coffee beans (ochratoxin A, 3.0 mg/kg) as a starting point for a regulatory reference value.

### 3.5.3. Thermal Reaction and Degradation Products, Including Aldehydes, Furans, Furfurals, and Acrylamide

Furfural and hydroxymethylfurfural (HMF) are organic compounds that result from the degradation of sugar. They consist of a furan ring and aldehyde compounds. Roasted date seed extract and liquid fraction contain both compounds, and their levels increase with higher temperatures [1]. The highest reference values and intake levels of furan are expected to be consumed through *C. arabica* coffee. Dry fruits and juices also contain high levels of furan. Currently, there are varying opinions on the toxicity of furan. Furan has been classified as possibly carcinogenic to humans (IARC group 2B), while HMF has shown no carcinogenicity in an in vivo rat study. On average, a person ingests 4 to 30 mg of HMF daily. Based on acute and subchronic animal data, a maximum dose of 80 to 100 mg/kg b.w. can be declared as the NOAEL, where no adverse effects are ex-

pected [70,71]. Nine different furans and furanones were detected in roasted date seeds, accounting for 5.9% to 26.5% of the total volatile mass. The most commonly detected compounds were 5-methyl-2-furaldehyde and 3-furfural [41]. Based on these findings, it can be concluded that the consumption of date seed coffee would not exceed the set NOAEL, as only minor amounts of furan and furanones are included in a cup of coffee after grinding and brewing.

During the roasting process, degradation products such as acrylamide (AA) or melanoidin are formed. As a result of the Maillard reaction and roasting temperatures above 120 °C, starch and sugars in reaction with amino acids are degraded to acrylamide. Previous studies have shown that acrylamide is probably carcinogenic to humans (IARC group 2A) [72,73]. It is known that *C. arabica* coffee contains a certain amount of acrylamide. An average of 678 µg/kg soluble coffee and 250 µg/kg roasted coffee powder was measured in an 8-year consumption study from 2007 to 2015. It has been shown that even high coffee consumption only accounts for 5% of the total AA intake [74]. Khaloo Kermani et al. investigated the potential of AA and melanoidin in coffee beverages made from roasted date seed powder due to the roasting process of date seeds [75]. Five different varieties of roasted date seeds were sampled and compared to *C. arabica* coffee using the HPLC/PDA method for analysis. The date seed coffee samples contained between 361 and 129 µg/kg AA (and in the coffee brew, 25 and 68 µg/L), which is significantly lower than the AA content in *C. arabica* coffee (1825 µg/kg, or brew: 346 µg/L). The melanoidin content was highest in the darker roasts and had a reversed relation with AA [75]. Compared to other study reports, the findings for date seed coffee and *C. arabica* coffee are relatively high. Mojrian Sharghi et al. conducted a study comparing a 100% *C. arabica* sample to a 100% date seed coffee sample, and three blends with different combinations (10%, 35%, and 50% date seed content) in between [60]. The AA content in the pure *C. arabica* coffee brew was about 5.1 µg/100 mL, and decreased in the mixture. The lowest AA content was measured with 1.7 µg/100 mL in the pure date seed brew [60]. The temperature at which coffee beans are roasted has an impact on the amount of AA present in the final product. Lachenmeier et al. conducted a study on the AA levels in different coffee roasts using the standard method EN 16618:2015. The lightest coffee roast had the highest AA content at 470 µg/kg, while the darker roasts had lower levels. The darkest blend had the lowest AA content at 130 µg/kg. This confirms the inverse relationship between the AA content and coffee roasting [76]. These results are consistent with those of an EFSA report, which found that light coffee roasts have a high acrylamide content because the AA content decreases as the roasting process continues [77].

In conclusion, the AA content of roasted date seeds is lower or comparable to that of common coffee products [60]. According to regulation (EU) 2017/2158, the AA content of coffee products must be lower than the benchmark level of 400 µg/kg [78]. Therefore, the literature data on acrylamide in date seed coffee do not suggest a health risk.

Data on various known contaminants in date seeds are available. Further studies, especially on the mycotoxin content of roasted date seed products, are still required. However, the available results are sufficient for an initial assessment. Overall, the findings in (roasted) date seeds are in line with comparable coffee products. Safety margins are available for these products, which can cover preliminary coffee beverages made from date seed powder. Further investigations would be of great interest, since there are no data available on the impurities from pesticides, mineral oils (MOSHs/MOAHs), nitrate compounds, or other production-related contaminants such as benzo[a]pyrene.

## 4. Consumption and Use of Date Seeds

### 4.1. Preparation of Date Seed Coffee

#### 4.1.1. Processing of Date Seeds

The process of preparing date seeds for coffee-like beverages is similar to that of making coffee beans. The first essential step is to remove the pulp from the seeds, followed by four additional steps in the preparation process, as follows: soaking, drying, grinding,

and hot water extraction to obtain a coffee brew [36]. The depulped date seeds are soaked in water and thoroughly cleaned. A fermentation stage can be added to remove any remaining mucilage and to enhance the flavor of date seed coffee beverages. After the drying process, which can take up to 8 h at a temperature of 80 °C, the coffee is roasted [4] (Figure 3).

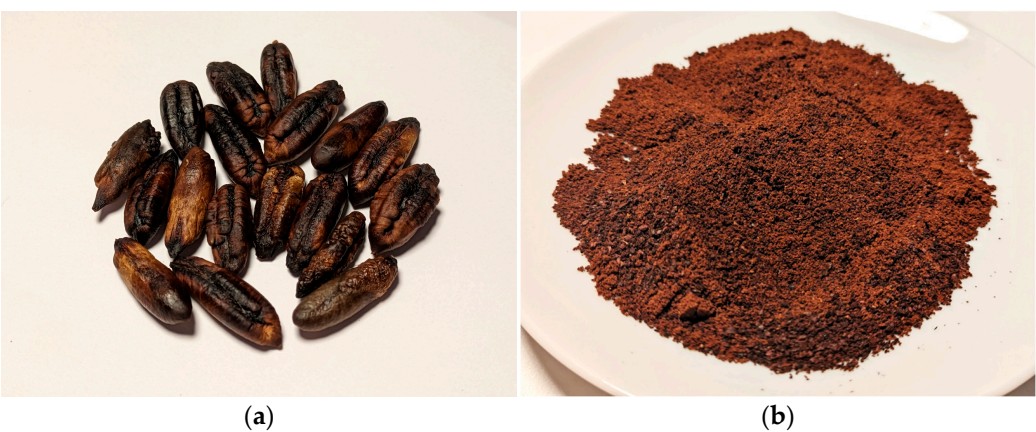

|  (a)  |  (b)  |

**Figure 3.** Macroscopic pictures (single treatment for illustrative purposes): (**a**) Date seeds, roasted for 20 min at 200 °C, in whole; (**b**) Date seed powder, roasted and grinded using a commercial coffee grinder for coffee preparation.

### 4.1.2. Roasting

The process of roasting coffee beans can be divided into three essential steps, including: drying, roasting (pyrolysis), and cooling [61]. The same three phases are also involved in the roasting of the date seeds. It is possible to use ordinary coffee roasting machines for the roasting of date seeds [58]. Ghnimi and Almansoori [4] tested three different date cultivars after roasting. The researchers roasted the seeds at 220 °C for 6 h, cooled them, and preserved them at 15 °C for further testing. It should be noted that the roasting time in this study is unusually high. According to [4], date seeds with a high density require higher roasting temperatures, as they are more resistant to heat. The exothermic phase during roasting can only be reached with a temperature above 145 °C [79]. During the roasting process, the weight of the date seeds decreases slightly, and their color changes over time to a darker, more brownish shade. Additionally, the pH level drops from 5.67 to 4.56, and the hardness and moisture content of the date seeds change [58]. An increase in titratable acidity was also observed. These properties are similar to those found in roasted coffee beans [4]. A slight change in seed weight occurred at temperatures as low as 60 °C [80]. The roasting process also caused a slight change in the composition of all of the macronutrients in the coffee bean, decreasing their concentration [61]. A temperature between 180 and 220 °C for approximately 20 min is suggested as a good guideline for roasting date seeds [44]. One study found that date seed powder roasted at 220 °C for 20 min had a better taste than samples roasted at lower temperatures (180–200 °C) or not roasted at all [24]. Another study tested temperatures between 160 and 200 °C and a roasting time of 10 to 30 min [58]. A roasting time of 21.5 min at 200 °C was found to be optimal for date seeds. Although sensory attributes such as aroma and taste are slightly better rated after a shorter roasting period of about 10 min [58], it is important to consider the overall impact of the roasting time on the final product.

### 4.1.3. Grinding and Filtering

Finally, the dried date seeds are ready for the grinding process. The degree of grinding determines the water absorption and permeability of the date seeds. Date seeds can be crushed by hand using a mortar and pestle, or a coffee mill can be used for a finer result. As the roasting process continues, the hardness and bulk density of the date seeds decrease, requiring less force for milling [58]. In the last step, water is poured over the powder

and the filtering process begins, separating the coffee grounds [12]. The filter process determines the solubility.

Date seed coffee can be prepared using various techniques. In the traditional method, the date seed powder is mixed with hot water. This traditional coffee preparation techniques is known as *qahwa* in Bahrain [81]. The grinded date seed powder is cooked in boiling water for 10–15 min with spices such as cardamon and saffron added to it. The hot coffee-like beverage is then served directly from the cooking utensils, with the sediment settled on the bottom of the pot [81].

After the coffee grounds have settled, the beverage is ready for consumption. Other types of preparation may also be used, resulting in a different composition.

### 4.2. Date Seed Beverages

A well-known traditional beverage in Bahrain is the noncaffeinated coffee *qahwa* [81]. Coffee-like beverages made from roasted date seed powder are rated lower in quality and have a milder taste in the sensory evaluation. These beverages exhibit sourness, bitterness, color, and coffee flavor, but the overall taste sensation is comparable to Arabic coffee [4]. Because the consumer's acceptance of date seed coffee is crucial, a study was conducted to compare a standard coffee-like drink containing 3 g, 6 g, and 9 g of date seed coffee powder with a fourth variant containing 10 g of *C. arabica* coffee powder. The lower dose date seed powder test drinks (3 g and 6 g) were rated worse, but the high dose was comparable or better than the *C. arabica* coffee beverage [63]. Roasted date seeds at 180 °C for 20 min achieved a general acceptability score of 2.4 out of 5 in a sensory evaluation, with 1 being the highest score. The untreated date seeds received the highest score for taste and odor, but the color was not favored by the testers [24].

Date seed coffee can be mixed with other spices to enhance the flavor and for possible health benefits. The taste is generally well received and considered quite enjoyable. Date seed powder can be stored for about 11 months at room temperature [34].

### 4.3. Target Consumers

Coffee beverages made from date seeds do not contain caffeine, which makes them suitable for a wide range of people [4,64]. Consumption of two to three cups of coffee per day, equivalent to 400 mg of caffeine, is generally considered safe and without causing harm to overall health. Acute caffeine intakes of up to 3 mg/kg b.w. per day are not a cause for concern. It is recommended that caffeine intake be restricted to lower levels than those recommend for adults in vulnerable groups such as children, adolescents, and pregnant women. The recommended caffeine intake of 200 mg should not be exceeded for these groups, especially for pregnant and lactating women, as it could harm the infant [82]. Date seed coffee beverages could serve as a substitute for regular coffee. As for other components of date seed coffee, such as heat degradation products, the recommended maximum intake of *C. arabica* coffee beverages could be used as a reference quantity.

### 4.4. Other Food Uses

The possibility of using date seed powder as a partial substitute partly for wheat flour has been investigated in several studies. Date seed flour contains more fat and fiber than wheat flour, which affects several sensory attributes of baked goods, such as flat bread or cookie dough. These attributes directly relate to the acceptance or rejection of the final product. Mixing the flour for sourdough production with pulverized date seeds at a ratio of one-tenth helps to diminish bread staling and increases the cell wall thickness. No significant differences in sensory attributes were detected when comparing "Barbari" flatbread with and without date seed powder [33]. Substituting 1–3% of bread dough with date seed fiber has a limited effect on the end product. Compared to the standard bread, the volume of the baked bread was lower, but the stability was higher, and the bread color and crumb were slightly darker and redder [83]. Date seeds contain a high amount of flavonoids and total polyphenolic content. By using a mixture of date seed powder

and wheat flour for the cookie dough, the final product showed an increase in flavonoids and polyphenolics compared to cookies made with wheat flour alone. This increase in antioxidant activity enhances the quality of the cookies [84].

Besides date seed powder, extracts, fatty acids, and oil can also be produced from date seeds [28]. The use of date seed oil was first described at the beginning of the 20th century [27]. Despite the fact that date seeds contain only a small amount of oil, the industry should be further pursued due to the high quantity available at little to no cost [40]. Date seed oil contains a high amount of oleic acid, which has a beneficial effect on the stability of the oil [1].

A recent study investigated the performance of date seed protein concentrate. The study found that high-intensity ultrasound treatment improved the functional properties of the protein concentrate, such as solubility, making it an interesting protein extract for the food industry [25].

Adding roasted date seed powder to ice cream increases its total fiber content. This functional by-product can also enhance the nutritional value of other dairy products by increasing their fiber intake and antioxidant effect. It also gives the final product a unique appearance due to the reddish color contained in the seeds [10].

### *4.5. Non-Food Use Possibilities*

In addition to food production, studies have explored other possibilities for reusing date seed waste. A recent study demonstrates that activated carbon can be produced from date seeds, which can be used for water treatment to remove waste and suspended matter [85]. The high polyphenolic content of date seeds may also be beneficial for the nutricosmetic and pharmaceutical industries [28]. Date seed oil can effectively protect human skin from oxidative damage, as well as from UV-B and UV-A radiation [1].

Furthermore, date seed waste can be used for the production of biodiesel [86] and as biomass for the bio-oil production. It is important not to exceed 450 °C, as the oil production only increases linearly until this temperature [87].

## 5. Toxicological Data

For many decades, waste products of the date industry, such as date seeds and other plant components, have been safely fed to animal livestock [31]. Acute toxicity data are available and an $LD_{50}$ has been determined. Although some chronic exposure and feeding studies of date seed coffee extracts have been carried out, no inhalation studies or dermal chronic toxicity tests have been performed.

### *5.1. Acute Toxicological Data*

Acute toxicity describes the adverse effects of a potent single or multiple doses of a substance administered orally or by inhalation on an organism over a short period of time not exceeding 24 h (Table 5).

**Table 5.** Overview of acute toxicity data of date seeds (DS).

| Test Objective | Animal Species | Duration (h) | Outcome | Reference |
|---|---|---|---|---|
| Acute toxicity, oral | mouse | 24 | $LD_{50}$ 6.75 g/kg b.w. for mice | [88] |
| Acute toxicity, oral | mouse | 24–72 | $LD_{50}$ of 1877 ± 39 mg/kg i.p. | [41] |

In their study, Mohamed and Al-Okbi [88] tested the acute toxicity of date seed extract on white, adult, female, and male albino mice using the Goodman et al. test method, as described in 1980. Date seeds were extracted using methyl alcohol. For sample preparation, the dry extracts were dissolved in distilled water and orally administered to mice within a 24 h time window. The dose was gradually increased over time. Once the dose reached

of 6 g/kg b.w., half of the test animals died. Following the method described by Paget and Barnes, an $LD_{50}$ was calculated, which was determined to be 6.75 g/kg b.w. for mice, which is equivalent to 52.4 g for a 70 kg male human [88]. Based on the results of this study, all test animals survived at a dosage of 4 g/kg b.w. and no adverse effects were reported, indicating a NOAEL (No Observed Adverse Effect Level). Another study confirms that date seeds have a safe margin. The $LD_{50}$ value was determined using the Litchfield and Wilcoxon method in seven groups of mice. Toxicity effects and mortality were observed after 24 h to 72 h. An $LD_{50}$ of $1877 \pm 39$ mg/kg i.p. was estimated, which is twice the $LD_{50}$ for *C. arabica* coffee ($733 \pm 39$ mg/kg i.p.) [41].

Overall, the use of date seeds as a coffee substitute is expected to have low acute toxicity. Therefore, it can be concluded that there is no acute risk to human health if the consumption of date seed coffee does not exceed the recommended daily intake of *C. arabica* coffee.

### 5.2. Subchronic and Chronic Toxicological Data

Several studies have been conducted on the chronic behavior of date seed extracts. Date seed extracts were administered at different dose levels. An overview of the available animal and human studies is given in Table 6.

**Table 6.** Overview of chronic animal and human studies using date seeds (DS).

| Effect Assessed | Species | Dose Rate | Duration (d) | Outcome | Reference |
|---|---|---|---|---|---|
| Effects of DS extract on liver function, lipid profile, and heart function | rat | Basal diet, supplemented by 15–20 mL/d DS extract | 56 | Increase in serum lipids; decrease in CK-NAC, CK-MB, LDH-P; high ALP; normal histological brain structure and mucosal lining of the stomach | [79] |
| Effects of DS extract on liver function and lipid profile | rat | (1) Feed: 1%, 5%, or 10% DS fiber (2) Feed: 5% or 10% DS fiber | 24 | Increase in serum lipids and cholesterol; overall weight gain | [29] |
| Effects of DS extract on the NO concentration, MDA level, total cholesterol, triglycerides (TGs), AST, ALT, amyloid A, and C-reactive protein (CRP) | rat | Daily oral dose of DS extract at 200 mg/kg b.w. | 70 | No effect in DS extract group versus control | [51] |
| Effects of DS extract in a high-fat diet on body and liver weight, glucose and insulin, total cholesterol, and TGs | rat | DS extracts of QT, BR, or RT variety; oral; 300 mg and 600 mg/kg | 56 | Improved hepatocytes and parenchymal structures, serum lipids, and lower body and liver weight | [89] |
| Effects of DS extract on the NO concentration, MDA level, total cholesterol, TGs, AST, ALT, creatine, urea, urid acid, troponin T, HbA1c, amyloid A, and CRP | human (55 female, BMI ~30) | DS coffee beverage, $7 \pm 0.5$ g/d (DS) | 70 | Decrease in NO, MDA, LDL, TGs, AST, ALT, amyloid A, CRP, HbA1c, creatine, urea, urid acid, troponin T, and a slight increase in HDL | [51] |

An in vivo rat study was performed to compare the physiological effects of *C. arabica* coffee, date seed coffee, *Cassia occidentalis* seed coffee, and barley coffee consumption [79]. All test samples were prepared identically. A group of 25 male adult rats were divided into five sub-groups. The control group received water only and the test groups received a basal diet of 15–20 mL per day of each coffee-like beverage over 8 weeks. There were no significant differences in body weight gain or organ weight among all groups. The group that consumed date seed coffee showed a normal histological brain structure and mucosal lining of the stomach. The serum lipid profile revealed the highest total lipid value in the date seed coffee group ($773 \pm 37$ mg/dL) compared to the control group ($617 \pm 30$ mg/dL). Total cholesterol, LDL, and HDL levels were slightly increased in the date seed coffee group, but were significantly lower than those in the *C. arabica* coffee sample group. Triglyceride levels were measured at $139 \pm 26$ mg/dL compared to $95 \pm 6$ mg/dL in the control group.

Consumption of date seed coffee resulted in a decrease in creatine kinase activity (CK-NAC and CK-MB), while LDH-P activity showed a slight increase. The group that consumed date seed coffee showed a significantly higher ALP value (111.9 µ/mL) compared to the control group (71.5 µ/mL) and a lower AST value, which was half of the control group's value. An increase in the ALP value may indicate adverse effects on liver cells and could be related to cholestasis, but the elevated reading should still be within the reference range. The study linked the increase to the degradation of fatty acids. However, the study did not provide any further description of possible harmful impacts or a histological examination of liver tissue [79]. A feeding study was conducted to investigate the effects of date seed fiber on male Sprague–Dawley rats [29]. The test groups received either 1%, 5%, or 10% date seed fiber (adolescent rats) over 24 days. In the second series of tests, adult rats were given either 2.5% or 15% date seed fiber in their food. The feed was provided ad libitum and the feed intake was not restricted. The study showed that the test animals consumed more food when the diet had a higher crude fiber content. At the conclusion of the experiment, the test animals were euthanized and their livers were removed for subsequent histological examination. The inclusion of date seed fiber in animal feed impacted the lipid metabolism of growing and adult male rats, leading to fatty infiltration in the liver at all stages of development. The livers of the test animals showed increased lipid deposits. The results of the feeding study confirmed an increase in serum lipids and cholesterol even at low dosages of date seed flour. Additionally, the overall weight gain was better than that of the control group, which received cellulose [29]. In another study, alternative treatments for diabetes were tested in rats [51]. Over a ten-week period, test animals (40 adult male Sprague–Dawley rats) were divided into diabetic and non-diabetic groups and given an oral dose of 200 mg/kg b.w. date seed extract daily, or an i.p. dose of alloxan monohydrate to treat diabetes. The test group consuming only the DS extract showed no deviations compared to the control group in NO, MDA, AST, ALT, amyloid A, CRP, and glucose, and only a slight increase in TGs, but no change in cholesterol levels, LDL, and HDL. Therefore, neither positive nor negative effects can be expected from daily oral consumption of date seed beverages. Interestingly, the diabetic test groups treated with date seed coffee drinks and alloxan monohydrate showed a decrease in NO, MDA, amyloid A, and CRP, as well as in AST and ALT, total cholesterol, TGs, LDL, but an increase in HDL compared to the diabetic group treated with alloxan monohydrate alone. But, overall, no harmful effects related to the consumption of date seed coffee have been shown [51]. Similar results were obtained in a 28-day study in adult male Wistar albino rats [89]. The rats were fed a high-fat diet ad libitum. The test groups were divided and received either medication with atorvastatin (for the treatment of dyslipidemia and the prevention of cardiovascular disease) or date seed extracts from different cultivars in two doses of 300 or 600 mg/kg b.w. rat daily. Over time, the high-fat diet resulted in increased body and liver weight, increased food intake, and higher glucose and insulin levels. Compared to the high-fat diet control group, the groups fed date seed extracts (from the Barhi and Ruthana cultivars) had lower liver and total body weights. Histological findings showed a decrease in the lipid deposition and a good hepatocyte and parenchymal structure. After repeated consumption of date seed extracts, leptin, AST and ALT, TGs, and total cholesterol were reduced, with an increase in HDL and a reduction in LDL in the blood concentration being confirmed, but no change in the glucose or insulin levels. Overall, no adverse effects were observed after the daily consumption of either 300 or 600 mg/kg b.w. date seed extract. Similar metabolic effects to the test substance atorvastatin were observed in the animals treated with the date seed extract [89].

Felemban and Hamouda [51] conducted parallel observations in human volunteers in addition to the diabetic animal study. The human study was conducted between 2016 and 2021 and included 55 female volunteers who were non-diabetic and with a BMI around 30, which can be considered overweight. The group was divided into two groups, and the second group received date seed coffee at a dose of $7 \pm 0.5$ g/day for at least 3 months. Group 1 was the control group and did not receive any test substance.

Group 2 showed a significant decrease in various serum markers, including NO (34.7%), MDA (43.9%), amyloid A (79.6%), CRP (60.9%), HbA1c (33.8%), and troponin T (16.4%) compared to group 1. This decrease was associated with a decrease in total cholesterol (14.2%), TGs (28.1%), LDL (6.1%), AST (8.8%), and ALT (11.6%) in group 2 compared to group 1. In addition, group 2 showed an 8.5% increase in HDL and a significant reduction in serum creatinine (16.4%), urea (45.4%), and uric acid (23%) compared to group 1. Diabetes markers (hba1c) improved to 4.1%, which is considered a non-diabetic condition, and inflammation and lipid peroxidation were reduced throughout the treatment period. Metabolic functions of several organs were improved (liver, kidney, and heart) [51].

### 5.3. Cytotoxic Activity

The cytotoxic effect of date seed samples was tested on MCF-7, breast cancer cells, and HCT-116, colon cancer cells and cell lines. Three repetitions were conducted, and the results showed an overall low cytotoxicity of the tested organic extracts [44]. Additionally, another study found that concentrations of up to 10 mg/mL of roasted date seed powder did not result in any inhibition of 50% MRC-5 cell viability. Therefore, it is unlikely that moderate consumption of date seed coffee would have toxic effects on human lung fibroblasts [35].

### 5.4. Other Animal Data

As previously mentioned, consuming date seeds may impact metabolic processes and have an antidiabetic effect due to their high fiber content and the presence of secondary plant metabolites. A study conducted in vitro using rat intestinal $\alpha$-glucosidase investigated the influence of bound polyphenols on starch digestion [55]. The polyphenolic compounds found in date seeds exhibited an inhibitory effect against rat intestinal $\alpha$-glucosidase. Simultaneously, the inhibitory effect on $\alpha$-amylase was low, which suggests potent antihyperglycemic properties [55].

Additionally, an in vivo study provided insight into the antioxidant and antiinflammatory effects of date fruit and date seed extracts against adjuvant arthritis [88]. The study utilized white male albino rats as test animals, which were divided into five groups. Three of these groups were administered daily doses of either date fruit methanolic or water extracts, as well as date seed methanolic extracts, via oral administration at a rate of 500 mg/kg rat b.w. for 14 consecutive days. At the beginning of the experiment, all test animals, except for one healthy control group, received an injection in the right hindfoot paw to induce the adjuvant arthritis. On day three of the experiment, the maximum inflammation was measured in foot thickness. Afterward, the swelling continuously decreased. The test groups treated with date fruit and seed extracts showed lesser inflammation in the paw. By the end of the study, the date seed extract group showed a reduction of 35.5% (date fruits: 61.3–67.8%). The antioxidant and antiinflammatory effects were confirmed [88].

Date seed coffee may have a positive impact on cognitive impairment caused by Alzheimer's disease (AD) [90]. In a study, four groups of adult male Wistar albino rats were treated with different combinations of aluminum chloride (10 mg/kg b.w., i.p.), *C. arabica* coffee, and date seed powder extracts (240 mg/kg b.w., i.p.) daily for a total of two weeks. The comparison of the time needed for the food search showed superior effects in the *C. arabica* coffee group in terms of cognitive function. On the other hand, the group of rats that consumed date seed coffee showed improved AD-modifying ability by reducing the $A_{\beta}$ protein levels. Possibly due to the high flavonoid content in date seeds, the consumption of C. *arabica* and date seed coffee led to a direct reduction in $A_{\beta}$ levels in the serum. Additionally, liver and kidney functional blood parameters showed an improvement compared to the group treated with only aluminum chloride. [90]. Therefore, further research should be conducted to investigate the potential of date seed coffee on the reduction in $A_{\beta}$ and the possibility of providing a treatment option for AD disease.

The poultry industry is actively seeking solutions to combat aflatoxicosis in poultry livestock, caused by the intake of aflatoxin-contaminated feed. This can lead to various health problems in the livestock. Typically, chemical mycotoxin binders such as HSCAS

(hydrated sodium calcium aluminosilicate) are added to the feed, but they also have adverse effects. In a study, the effectiveness of adding date seeds to the basal diet was compared to that of HSCAS. Aflatoxin was added at a rate of 100 μg/kg basal diet for 35 days. The test groups showed damage to the liver and kidney, including histopathological changes, interstitial nephritis, and mononuclear cell infiltrations. Both groups fed with date seeds (2% and 4%, respectively) demonstrated a positive effect against the damage caused by aflatoxin comparable to or better than HSCAS. The best results were achieved with 4% date seeds mixed into the basal diet [91].

## 6. Conclusions and Future Trends

Date seeds have a long history of use. In Europe, the first records of date seeds being used as coffee date back to the 19th century. Date seeds make up about 10–15% of the total fruit weight and are mainly composed of carbohydrates. Dates are high in fiber and low in sugar content, with a fat and protein content of around 5 to 10%. The fatty acids in date seeds have good oxidative stability and a high content of unsaturated acids. The composition of date seeds can be influenced by the agroclimatic conditions, cultivar, and food processing techniques, such as heating and roasting, which can also affect the mineral content. It is important to monitor metals that can be absorbed via the water supply. However, based on the analyzed data, it is unlikely that there will be an exceedance of daily intake levels or adverse health effects from consuming date seed coffee. Date seeds are rich in phenols and flavonoids, as well as vitamin E and carotenoids. Several studies have confirmed that date seeds do not contain caffeine. The high concentration of beneficial ingredients and their antioxidant effects make them a valuable ingredient for food production.

Mycotoxins have been found to contaminate fruits, and manufacturing processes can influence their levels. Although date seeds are less contaminated than its pulp, it is essential to monitor and limit the risk of exposure. It is known that mycotoxins have a genotoxic effect on organisms. To assess date seed coffee, the regulatory maximum levels for ochratoxin A (3.0 mg/kg) defined for coffee beans can be used, as exposure levels to both beverages are expected to be similar. Heat degradation products were detected in roasted date seeds. The content measured in date seeds is lower than that in *C. arabica* coffee beans. Compared to the existing benchmark value for coffee beans, no health risk is expected with moderate consumption.

Optimum roasting results can be achieved by maintaining a heat of 200 °C for about 20 min. Date seed coffee can be consumed pure or mixed with spices or *C. arabica* coffee. Several techniques can be used to prepare coffee beverages. The absence of caffeine can be seen as an advantage for certain target groups.

Meaningful studies have already been carried out regarding toxicological data. Acute toxicity has been determined, and two studies provided insight into the toxicity effects and mortality after 24 h (to 72 h). An LD$_{50}$ for mice was also determined. Both values exceed the reference values for *C. arabica* coffee, indicating a low acute risk for date seed coffee. Subchronic and chronic toxicological data, mainly tested in rodents, as well as data from a human study with 55 participants, are available. The tested dose rates cover a possible daily intake of three average cups of coffee and the required amount of date seed coffee powder. No negative effects were reported. No adverse changes were observed in the organ metabolism or histological examination of the respective organs. Rodent and human studies have shown improved blood lipid values after repeated consumption of date seed extracts. However, there are no available studies on inhalation or dermal toxicity, only oral data. Additionally, testing on the cell lines MRC-5, MCF-7, and HCT-116 did not reveal any cytotoxic effects.

In conclusion, there is room for further investigation. While several animal studies have provided a good overview of subchronic toxicity, only one human study is available. The test group was limited to female volunteers, but the duration was at least 90 days or even longer, and information on blood chemistry and inflammatory markers was provided

throughout. However, the available animal studies do not meet all OECD criteria, but no adverse effects on animal health were reported. The studies tested the health effects after the consumption of date seed fiber and date seed extract, which would be related to the consumption of coffee beverages.

No cytotoxic effects were reported. A standard test required for genotoxicity testing has not yet been performed. Genotoxicity testing is considered a fundamental component of the risk assessment, but no genotoxicity data accepted by the OECD requirements are available. Therefore, this has to be marked as a missing data requirement to finally prove the safety of the product, especially if novel food guidelines for risk assessment would be demanded. Carcinogenicity and reproductive toxicity studies of date seeds are also not available. No evidence of carcinogenicity or reproductive toxicity has been documented in subchronic studies. Finally, date seeds have been fed to livestock for many years and no adverse effects have been reported, so consideration should be given to whether carcinogenicity or reproductive toxicity studies are needed and justified. Tests on immunotoxicity, endocrine activity, or allergenic potential are missing. Since allergenic effects are often related to the protein content of the food consumed, the risk of date seeds triggering allergenicity should be very low due to their low protein content. Date seed oil is now being used in cosmetics and there have been no reports of allergic reactions [1,28]. With increasing use in the cosmetic sector and food industry, further toxicologically relevant tests may be performed. But for the time being, it can be concluded that the daily cup of date seed coffee does not appear to pose a health risk beyond the one of "normal" coffee.

**Author Contributions:** Conceptualization, R.K. and D.W.L.; methodology, R.K.; formal analysis, R.K.; investigation, R.K.; writing—original draft preparation, R.K.; writing—review and editing, D.W.L. and H.F.; visualization, R.K.; supervision, D.W.L.; project administration, D.W.L. All authors have read and agreed to the published version of the manuscript.

**Funding:** This research received no external funding.

**Institutional Review Board Statement:** Not applicable.

**Informed Consent Statement:** Not applicable.

**Data Availability Statement:** No new data have been generated. Data sharing is not applicable to this article.

**Acknowledgments:** This manuscript was published as part of the Postgraduate Study of "Toxicology and Environmental Protection" at the University of Leipzig, Germany.

**Conflicts of Interest:** The authors declare no conflicts of interest.

## Abbreviations

| | |
|---|---|
| AA | Acrylamide |
| $A_\beta$ | Beta amyloid |
| AD | Alzheimer's disease |
| ALP | Alkaline phosphatase |
| ALT | Alanine transaminase |
| AST | Aspartate transaminase |
| b.w. | Body weight |
| CK | Creatine kinase |
| CRP | C-reactive protein |
| DS | Date seeds |
| d.w. | Dry weight |
| EFSA | European Food Safety Authority |
| EU | European Union |
| GAE | Gallic acid equivalent |
| GC | Gas chromatography |
| $HbA_{1c}$ | Glycated hemoglobin |
| HCT-116 | Colon cancer cells |

| | |
|---|---|
| HDL | High-density lipoprotein |
| HMF | Hydroxymethylfurfural |
| HPLC | High-performance liquid chromatography |
| HSCAS | Hydrated sodium calcium aluminosilicate |
| IARC | International Agency for Research on Cancer |
| i.p. | Intraperitoneal |
| μg | Microgram, corresponding to $10^{-3}$ mg |
| LC | Liquid chromatography |
| $LD_{50}$ | Lethal dose, 50% |
| LDH-P | Lactate dehydrogenase |
| LDL | Low-density lipoprotein |
| MCF-7 | Breast cancer cells |
| MDA | Malondialdehyde |
| mg | Milligram |
| mL | Milliliter |
| MOAHs | Mineral oil aromatic hydrocarbons |
| MOSHs | Mineral oil saturated hydrocarbons |
| MRC-5 | Human embryonal lung fibroblast |
| MS | Mass spectrometry |
| n.a. | Not applicable |
| n.d. | Not detected |
| NO | Nitric oxide |
| NOAEL | No Observed Adverse Effect Level |
| ORAC | Oxygen radical absorbance capacity |
| PDA | Photodiode array |
| $SC\text{-}CO_2$ | Supercritical carbon dioxide (for supercritical fluid extraction (SFE)) |
| sp., spp. | Biology, undefined species |
| TGs | Triglycerides, total cholesterol |
| TPC | Total phenolic content |
| TWI | Tolerable weekly intake |
| UAE | Ultrasonic-assisted extraction |
| UV | Ultraviolet |

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
