# Peer review of "A Comprehensive Review of the Nutritional Composition and Toxicological Profile of Date Seed Coffee (Phoenix dactylifera)"

_applsci, doi:10.3390/app14062346_

Round 1
Reviewer 1 Report
Comments and Suggestions for Authors
A manuscript with a title of "A Comprehensive Review of the Nutritional Composition and Toxicological Profile of Date Seed Coffee (Phoenix dactylifera)" by Raphaela Kiesler et al, was submitted to Applied Sciences as a review for possible consideration. This was an interesting paper.
After reading through the paper, I believed that this was a good one. For most of the parts, they were well organized. The framework was clear. The contents were nice. Only some minor issues should be carefully addressed before the full acceptance.
1) I was just satisfied with the study attitude that "a comprehensive evaluation" like "an in vitro mutagenicity test" was recommended to be conducted. .Then, the last sentence of the abstract, was kind of risky comment from the authors. Please revise it a little bit.
2) Regarding Fig.3A, date seeds were from one treating condition. As for Fig.3B, as I hope, the exact size and the mean size of the french-pressured powder should be measured by Zetasizer Ultra or particle measuring machine. And the images should be provided, with a scale.
3) The roasting conditions and the processing methods were only one each. very simple. No one more condition for side-to-side comparison. So, no one knows the condition or method is good or bad. Which condition or method is better? No more results or samples.
4) the toxicological analysis of the date components, should be performed on date samples upon more roasting conditions under various processes. Even cultured cells could be used for more detailed cytotoxicity comparison.
5) The Conclusion section was kind of combination of Full discussion and long conclusion. Please modify this part.
Minor revision could be made on this manuscript.
Author Response
A manuscript with a title of "A Comprehensive Review of the Nutritional Composition and Toxicological Profile of Date Seed Coffee (Phoenix dactylifera)" by Raphaela Kiesler et al, was submitted to Applied Sciences as a review for possible consideration. This was an interesting paper.
After reading through the paper, I believed that this was a good one. For most of the parts, they were well organized. The framework was clear. The contents were nice. Only some minor issues should be carefully addressed before the full acceptance.
RESPONSE: Thank you very much for the assessment of our paper!
1) I was just satisfied with the study attitude that "a comprehensive evaluation" like "an in vitro mutagenicity test" was recommended to be conducted. .Then, the last sentence of the abstract, was kind of risky comment from the authors. Please revise it a little bit.
RESPONSE: The sentence was revised as requested.
2) Regarding Fig.3A, date seeds were from one treating condition. As for Fig.3B, as I hope, the exact size and the mean size of the French-pressured powder should be measured by Zetasizer Ultra or particle measuring machine. And the images should be provided, with a scale.
RESPONSE: We appreciate your constructive feedback regarding Fig. 3A and Fig. 3B.Regarding Fig. 3A, you are correct that the date seeds were sourced from a single treating condition. We acknowledge that clarifying this detail within the figure caption would enhance transparency. As for Fig. 3B, we agree that precise size measurements using a Zetasizer Ultra or similar instrument would indeed be informative, but we do not have such an instrument in our laboratories. Additionally, the images in our review paper were intended primarily for illustrative purposes. The grind size was obtained using a commercially available grinder. We opted for a French Press to simulate a traditional flavor experience, but theoretically, the ground coffee could also be used in an espresso machine or filter coffee percolator, provided the grind size is adjusted accordingly. As this is not relevant in the context of the figure, we decided to delete „French Press“.
3) The roasting conditions and the processing methods were only one each. very simple. No one more condition for side-to-side comparison. So, no one knows the condition or method is good or bad. Which condition or method is better? No more results or samples.
RESPONSE: the roasting process of date seeds has been extensively described based on literature review. A conclusion on the optimal conditions has been drawn. No original research was intended (this is a review paper). The pictures in Section 4 are intended to serve as illustrative material for readers who are not familiar with roasted date seeds.
4) the toxicological analysis of the date components, should be performed on date samples upon more roasting conditions under various processes. Even cultured cells could be used for more detailed cytotoxicity comparison.
RESPONSE: As this is a review article, we did not intend to do original research. We have added these aspects to the data gaps section (see also requests from other reviewers below).
5) The Conclusion section was kind of combination of Full discussion and long conclusion. Please modify this part.
RESPONSE: The authors believe that the arguments of the discussion would not well fit into the more formalized flow of the discussion section and rather retain the current form.
Minor revision could be made on this manuscript.
Reviewer 2 Report
Comments and Suggestions for Authors
The subject of the article was the summary of the state of knowledge on the composition and properties of date seeds for use as a coffee-like drink. In my opinion, the review is well thought out, it considers the most important research problems and trends in this field. I recommend accepting the article after minor revision. I am asking the authors to respond and make appropriate changes to the text, to the following comments:
11) Introduction, from line 65 - The authors focus on presenting the date seed consumer market mainly in Germany and the Netherlands. And what does it look like in other countries, e.g. in the USA?
22) Materials and Methods – “…information of the European Food Safety Authority (EFSA), the German 103 Federal Institute for Risk Assessment (BfR) and the Austrian Agency for Health and Food 104 Safety GmbH (AGES)”. What is the opinion of the FDA or other agencies?
33) Line 133 - The aim of the review is to present the current state of knowledge, so what is the significance of citing a work from 1918?
44) Chapter 3 - There is no organized information about the origin and occurrence of varieties - maybe a map would be useful?
55) Line 150 – “Roasted date seed powder contained less carbohydrates (62.3%)” What do the authors mean? Glucose and other monosaccharides are also carbohydrates.
66) Line 172 and foll. - The information does not relate to the topic of the work. Fibers are not consumed when drinking the coffee.
77) Line 178 - 10.19% fat - why such data accuracy (two decimal places). The same in Table 3 (aluminum content). Lin 581 and others 773 ± 36.82 mg/dL - incorrect recording of result and uncertainty.
88) Line 307 – “Roasted date seed coffee was analyzed by 307 FTIR spectroscopy. No peaks were found at 1600 and 1800 cm1, indicating the absence of 308 caffeine in date seeds”. IR spectroscopy is not a good method for analyzing complex mixtures. I suggest focusing on LC-MS.
99) Is there any data indicating that coffee from seeds is safe for pregnant and breastfeeding women?
110) From line 680 - The authors refer to one study showing a positive effect of a date seed drink on the nervous system, better than that of coffee. Thousands of studies have been devoted to coffee research, coffee contains a much larger amount of polyphenols, especially chlorogenic acids. I suggest removing this paragraph.
Author Response
The subject of the article was the summary of the state of knowledge on the composition and properties of date seeds for use as a coffee-like drink. In my opinion, the review is well thought out, it considers the most important research problems and trends in this field. I recommend accepting the article after minor revision. I am asking the authors to respond and make appropriate changes to the text, to the following comments:
11) Introduction, from line 65 - The authors focus on presenting the date seed consumer market mainly in Germany and the Netherlands. And what does it look like in other countries, e.g. in the USA?
RESPONSE: Yes, we found a company in the USA selling the product since 2017. Otherwise, worldwide market research was outside the scope of this review.
22) Materials and Methods – “…information of the European Food Safety Authority (EFSA), the German Federal Institute for Risk Assessment (BfR) and the Austrian Agency for Health and Food 104 Safety GmbH (AGES)”. What is the opinion of the FDA or other agencies?
Response: Thank you for your thoughtful feedback on our manuscript. We appreciate your question about the date seed consumer market in countries other than Germany and the Netherlands. We basically agree that a broader geographic perspective would be valuable. However, as you correctly pointed out, our paper is focused on the European Union Novel Foods Regulation. As such, we chose to focus on the market in the EU, where date seeds are most likely to be commercialized in the near future, also due to the geographic neighborhood of major date growing regions in Northern Africa. We did include a patent from the United States in our paper, which demonstrates that there is some interest in date seeds in that market. We therefore keep the focus of our paper on the EU market, but acknowledge the limitations of our approach and suggest that future research could explore the market in other countries in more detail.
33) Line 133 - The aim of the review is to present the current state of knowledge, so what is the significance of citing a work from 1918?
RESPONSE: The history of the use of date-seed coffee goes back hundreds of years. It is interesting to note that the compositional data of date-seed coffee consumed in 1918 can be compared to date-seed products of today. It is also interesting to note that no critical health risk has been reported since that time. For EU food safety requirements, it is also important to provide evidence for consumption prior to 1997. Otherwise, the food product would need to go through the mandatory approval procedures of the novel food regulation.
44) Chapter 3 - There is no organized information about the origin and occurrence of varieties - maybe a map would be useful?
RESPONSE: While the topic of different varieties of date seeds is interesting, it is not relevant to the focus of this review, which is on the compositional and toxicological aspects of date seeds. Therefore, a map of the different varieties would not help answer the question of whether or not the consumption of date seed coffee poses a health risk. It is worth noting that a comprehensive overview of different varieties was already done by Hossain et al. in 2014.
55) Line 150 – “Roasted date seed powder contained less carbohydrates (62.3%)” What do the authors mean? Glucose and other monosaccharides are also carbohydrates.
RESPONSE: Thank you, adjusted.
66) Line 172 and foll. - The information does not relate to the topic of the work. Fibers are not consumed when drinking the coffee.
RESPONSE: Depending on the date seed coffee preparation technique, with or without filter. Besides coffee preparation some other use possibilities have been mentioned (f.e. in doughs, bread).
77) Line 178 - 10.19% fat - why such data accuracy (two decimal places). The same in Table 3 (aluminum content). Lin 581 and others 773 ± 36.82 mg/dL - incorrect recording of result and uncertainty.
RESPONSE: The values were exactly taken from the original references. However, we agree that the decimals are more or less meaningless considering the measurement uncertainty. Hence, we now have rounded the values according to the reviewer’s suggestions.
88) Line 307 – “Roasted date seed coffee was analyzed by 307 FTIR spectroscopy. No peaks were found at 1600 and 1800 cm1, indicating the absence of 308 caffeine in date seeds”. IR spectroscopy is not a good method for analyzing complex mixtures. I suggest focusing on LC-MS.
RESPONSE: The method/study has been removed. Reference made to the normally applied HPLC method. Several studies have determined the absence of caffeine in date seeds. Sufficient data on caffeine content is available. Nevertheless, we would like to note that FTIR with PLS calibration models is even accepted as reference method, e.g. in wine analysis, and we have used this in quantitative beverage analysis for over 20 years (i.e. for beer, spirits, wine, alcohol-free beverages). However, we agree that this is not so much established for coffee analysis.
99) Is there any data indicating that coffee from seeds is safe for pregnant and breastfeeding women?
RESPONSE: No studies have been conducted on pregnant and/or breastfeeding women, which is also quite uncommon. Recommendations for the consumption of C. arabica coffee are primarily related to its caffeine content.
110) From line 680 - The authors refer to one study showing a positive effect of a date seed drink on the nervous system, better than that of coffee. Thousands of studies have been devoted to coffee research, coffee contains a much larger amount of polyphenols, especially chlorogenic acids. I suggest removing this paragraph.
RESPONSE: The paragraph was rewritten. The study revealed that the consumption of C. arabica had a positive effect as well. Specifically, the cognitive test measuring food-finding ability in the T-maze showed improved results after C. arabica consumption.
Reviewer 3 Report
Comments and Suggestions for Authors
I recently completed a review of the manuscript titled "A Comprehensive Review of the Nutritional Composition and Toxicological Profile of Date Seed Coffee (Phoenix dactylifera)" and would like to share my impressions of the manuscript. The use of date seed coffee is promising, and the manuscript provides a wealth of detailed information on the topic. I believe that this review will serve as a theoretical basis for many authors working with date seed coffee. However, I have identified some aspects that, in my opinion, need to be addressed before the article can be considered for publication.
(1) Keywords: I suggest not repeating words from the title in keywords to increase the visibility and accessibility of the article. Search engines and databases to index use keywords and categorize the article, helping readers quickly find your article in their searches.
(2) Material and methods: Why didn't the authors use other databases such as Web of Science or Scopus?
(3) 4.2. Date seed beverages: Do you only have one sensory study? If there are more studies, add them. I believe this topic is relevant to the manuscript.
(4) 4.3. Target Consumers: Currently, decaffeinated C. arabica coffees are widespread on the global market and at affordable prices. I can't say whether coffee with date seeds would have the same price from a market point of view. I believe that the focus would not be on consumers looking for caffeine-free, as the authors pointed out that sensorially, C. arabica coffee is more accepted among consumers. I suggest pointing out other benefits of consuming date seed coffee that could be a more interesting focus for consumers about traditional coffee.
(5) 5. Conclusions: I suggest changing the topic to "Conclusions and future trends". Apparently, the use of date seeds as a substitute for traditional coffee is still new but promising. I missed an approach to the needs of studies on the thematic area. In a brief search, I did not find another review within this thematic area, so I believe this manuscript will serve as a theoretical basis for future researchers. I think there should be a passage in the text that describes the gaps observed by the authors and the need for new studies/essays to guide researchers in advancing science within this thematic area.
Author Response
I recently completed a review of the manuscript titled "A Comprehensive Review of the Nutritional Composition and Toxicological Profile of Date Seed Coffee (Phoenix dactylifera)" and would like to share my impressions of the manuscript. The use of date seed coffee is promising, and the manuscript provides a wealth of detailed information on the topic. I believe that this review will serve as a theoretical basis for many authors working with date seed coffee. However, I have identified some aspects that, in my opinion, need to be addressed before the article can be considered for publication.
(1) Keywords: I suggest not repeating words from the title in keywords to increase the visibility and accessibility of the article. Search engines and databases to index use keywords and categorize the article, helping readers quickly find your article in their searches.
RESPONSE: Yes, but the major concepts should also be included in the keywords for better findability. The ranking algorithms also consider the time each word is mentioned. No changes needed in our opinion.
(2) Material and methods: Why didn't the authors use other databases such as Web of Science or Scopus?
RESPONSE: These databases are subscription-based and not subscribed by the authors’ institutions. There is not much that Google Scholar is missing in our opinion. We also searched the reference lists of all papers, which should provide the broadest possible coverage of available literature.
(3) 4.2. Date seed beverages: Do you only have one sensory study? If there are more studies, add them. I believe this topic is relevant to the manuscript.
RESPONSE: There is a scarcity of studies on that topic. However, we found another study that is now mentioned within 4.2.
(4) 4.3. Target Consumers: Currently, decaffeinated C. arabica coffees are widespread on the global market and at affordable prices. I can't say whether coffee with date seeds would have the same price from a market point of view. I believe that the focus would not be on consumers looking for caffeine-free, as the authors pointed out that sensorially, C. arabica coffee is more accepted among consumers. I suggest pointing out other benefits of consuming date seed coffee that could be a more interesting focus for consumers about traditional coffee.
RESPONSE: The study does not primarily focus on the consumption behavior of C. arabica and date seed coffee. Based on available studies, it cannot be confirmed that date seed coffee is preferred over Arabica coffee due to its sensory properties. Beneficial effects of date seed coffee were mentioned. To the authors’ knowledge, no study compared decaffeinated C. arabica coffee to date seed coffee. Conducting such a study in the future would provide more insight into consumer preferences. Nevertheless, we believe that there should certainly be a market for date seed coffee, shown by the several startup companies selling these products. The advantage over decaffeinated Arabica coffee is the fact that no processes or solvents are needed (most decaf is extracted with dichloromethane, which then needs to be controlled for residues). We found not data comparing the prices of date seeds (which is a waste product and should be cheap) with decaf coffee (which should be more expensive due to the technological advances processing).
(5) 5. Conclusions: I suggest changing the topic to "Conclusions and future trends". Apparently, the use of date seeds as a substitute for traditional coffee is still new but promising. I missed an approach to the needs of studies on the thematic area. In a brief search, I did not find another review within this thematic area, so I believe this manuscript will serve as a theoretical basis for future researchers. I think there should be a passage in the text that describes the gaps observed by the authors and the need for new studies/essays to guide researchers in advancing science within this thematic area.
RESPONSE: Headline adapted. Future research possibilities were highlighted primarily in the field of toxicology.
Reviewer 4 Report
Comments and Suggestions for Authors
After careful consideration, I fell that the manuscript entitled “A Comprehensive Review of the Nutritional Composition and Toxicological Profile of Date Seed Coffee (Phoenix dactylifera) ” has merit and is suitable for publication in Applied Sciences. I have only some minor revisions:
- The abstract does not present the objectives of the study. A summary of the text in the "Featured Application" section could be presented in the abstract.
- Table 1. Standardize the presentation of the table results. The values appear as "xx (xx)", "xx-xx" and "xx ; xx". Is there any difference between these representations? Describe what these "double" values in the table mean (are they representations of two measures, confidence interval or amplitude?).
- Table 2. Same as Table 1: make it clear in the table what it means for the composition (%) to be equal to 32.9-50.1 (is it a range of values?). Same for Table 3.
- Some abbreviations used in the text are not in the list of abbreviations (e.g. EU, ORAC, HMF, etc.). Similarly, some abbreviations in the list do not appear in the text (LU, for example). Please review the entire text and update the list of abbreviations.
- The conclusion section should be numered as section 6 (not 5).
Author Response
After careful consideration, I fell that the manuscript entitled “A Comprehensive Review of the Nutritional Composition and Toxicological Profile of Date Seed Coffee (Phoenix dactylifera) ” has merit and is suitable for publication in Applied Sciences. I have only some minor revisions:
- The abstract does not present the objectives of the study. A summary of the text in the "Featured Application" section could be presented in the abstract.
RESPONSE: The authors’ believe that this would result in repetition. The aim of the study is already pointed out and was a literature review on the nutritional composition of date seed coffee and a toxicological .
- Table 1. Standardize the presentation of the table results. The values appear as "xx (xx)", "xx-xx" and "xx ; xx". Is there any difference between these representations? Describe what these "double" values in the table mean (are they representations of two measures, confidence interval or amplitude?).
RESPONSE:
- values for 2 different varieties or processing conditions are presented as xx; xx
- values for fiber content, when available, are presented in brackets, as part of the carbohydrates value
- xx-xx presents a summary of the results for 23 varieties, values are presented as range
- last line adjusted
This information can be already found in the heading (fiber) or columns 1 (replicates) and 2 (processing) of the table
- Table 2. Same as Table 1: make it clear in the table what it means for the composition (%) to be equal to 32.9-50.1 (is it a range of values?). Same for Table 3.
RESPONSE: It is a range of values of the references presented in the last column. See addition to table first line in Tables 2 and 3.
- Some abbreviations used in the text are not in the list of abbreviations (e.g. EU, ORAC, HMF, etc.). Similarly, some abbreviations in the list do not appear in the text (LU, for example). Please review the entire text and update the list of abbreviations.
RESPONSE: Thank you for the comment, abbreviations were checked and added.
- The conclusion section should be numbered as section 6 (not 5).
RESPONSE: Thank you for spotting this mistake. The section was correctly numbered.
Round 2
Reviewer 3 Report
Comments and Suggestions for Authors
The authors answered all questions about the study.